# NLRP3-Mediated Piezo1 Upregulation in ACC Inhibitory Parvalbumin-Expressing Interneurons Is Involved in Pain Processing after Peripheral Nerve Injury

**DOI:** 10.3390/ijms232113035

**Published:** 2022-10-27

**Authors:** Qiao-Yun Li, Yi-Wen Duan, Yao-Hui Zhou, Shao-Xia Chen, Yong-Yong Li, Ying Zang

**Affiliations:** 1Pain Research Center and Department of Physiology, Zhongshan Medical School of Sun Yat-sen University, 74 Zhongshan Road. 2, Guangzhou 510080, China; 2Department of Anesthesiology, Sun Yat-sen University Cancer Center, State Key Laboratory of Oncology in South China, Collaborative Innovation Center for Cancer Medicine, 651 Dongfeng Road East, Guangzhou 510060, China

**Keywords:** neuropathic pain, anterior cingulate cortex, Piezo1, parvalbumin interneuron, NLRP3

## Abstract

The anterior cingulate cortex (ACC) is particularly critical for pain information processing. Peripheral nerve injury triggers neuronal hyper-excitability in the ACC and mediates descending facilitation to the spinal dorsal horn. The mechanically gated ion channel Piezo1 is involved in the transmission of pain information in the peripheral nervous system. However, the pain-processing role of Piezo1 in the brain is unknown. In this work, we found that spared (sciatic) nerve injury (SNI) increased Piezo1 protein levels in inhibitory parvalbumin (PV)-expressing interneurons (PV-INs) but not in glutaminergic CaMKⅡ^+^ neurons, in the bilateral ACC. A reduction in the number of PV-INs but not in the number of CaMKⅡ^+^ neurons and a significant reduction in inhibitory synaptic terminals was observed in the SNI chronic pain model. Further, observation of morphological changes in the microglia in the ACC showed their activated amoeba-like transformation, with a reduction in process length and an increase in cell body area. Combined with the encapsulation of Piezo1-positive neurons by Iba1^+^ microglia, the loss of PV-INs after SNI might result from phagocytosis by the microglia. In cellular experiments, administration of recombinant rat TNF-α (rrTNF) to the BV2 cell culture or ACC neuron primary culture elevated the protein levels of Piezo1 and NOD-like receptor (NLR) family pyrin domain containing 3 (NLRP3). The administration of the NLRP3 inhibitor MCC950 in these cells blocked the rrTNF-induced expression of caspase-1 and interleukin-1β (key downstream factors of the activated NLRP3 inflammasome) in vitro and reversed the SNI-induced Piezo1 overexpression in the ACC and alleviated SNI-induced allodynia in vivo. These results suggest that NLRP3 may be the key factor in causing Piezo1 upregulation in SNI, promoting an imbalance between ACC excitation and inhibition by inducing the microglial phagocytosis of PV-INs and, thereby, facilitating spinal pain transmission.

## 1. Introduction

One of the big events in the field of neuroscience in 2021 was the awarding of the Nobel Prize in Physiology or Medicine to those responsible for the research on receptor channels responsible for the perception of touch (Piezo) and temperature (TRPV1) [1]. The evolutionarily conserved Piezo protein family comprises two mechanosensitive, non-selective cation channel proteins with relatively homologous structures, called Piezo1 and Piezo2 [2,3,4,5,6]. In the central nervous system (CNS), the mechanically gated Piezo ion channels are important regulators of neural stem cell differentiation [7], cell migration [8,9], axon guidance [10], and axonal myelination by oligodendrocytes (myelin-forming cells) [11]. In addition, Piezo channels are also involved in the transmission of pain information; however, current studies on Piezo have focused on its involvement in primary sensory pathways [12,13,14]. Piezo1 is selectively expressed in smaller dorsal root ganglion (DRG) neurons and mediates both mechanical pain and pain allergy [12], while Piezo2 is expressed in DRG neurons of all diameters and mediates a variety of sensory information, including that regarding proprioception and touch [14,15,16]. Although Piezo2 has no [14] or an inhibitory [17] or a partial role [18] in mechanical pain, its roles in tactile allodynia [14,18,19] and visceral pain [20] are crucial. In addition, both Piezo1 and Piezo2 are highly expressed in the trigeminal ganglion [21], mediating migraines (characterized by allodynia and mechanical hyperalgesia [21,22]) and trigeminal neuralgia (characterized by severe paroxysmal pain [23]).

Notably, although Piezo2 protein expression exists in the vast majority of DRG neurons, its expression is limited to a selection of neuronal types in specific brain regions, including neocortical and hippocampal pyramidal neurons, cerebellar Purkinje cells, and olfactory bulb mitral cells [24]. Therefore, the present study has explored the pain-processing role of Piezo1 in the brain. Piezo1 can be activated by mechanical perturbations of the membrane lipid bilayer [25], which cause Ca^2+^ influx as well as Ca^2+^ release from intracellular stores [26,27,28], leading to the activation of intracellular signaling pathways. In the CNS, Piezo1 is expressed preferentially on cortical myelinated neurons compared to oligodendrocytes [11]. The Piezo1 agonist Yoda1 [29] induces neuronal demyelination, while the Piezo1 inhibitor, Grammostola spatulata mechanotoxin-4 (GsMTx4) [30] has a neuroprotective effect and blocks chemically induced astrocyte toxicity and microglial activation [11]; these findings suggest a potential proinflammatory role of Piezo1.

The anterior cingulate cortex (ACC) is particularly critical for pain perception, pain aversion as well as the social transfer of pain [31,32,33,34], and its synapses can form “pain memory” [35]. In recent years, others’ [36,37,38,39,40] and our own research [34] have found that neuro-immune dysregulation in the ACC plays an important role in pain information processing. Peripheral nerve injury triggers neuronal hyper-excitability in the ACC superficial layer (Ⅱ/Ⅲ) [41,42,43]. Our recent work revealed that the increased expression of the Fractalkine/C-X3-C motif chemokine ligand 1 (CX3CL1) in the ACC following spared (sciatic) nerve injury (SNI), a unilateral chronic pain model [34,44], mediated descending facilitation to the spinal dorsal horn (SDH), and enhanced spinal neuroinflammation [45]. Therefore, an in-depth understanding of the molecular mechanism underlying neuroimmunity in the ACC following peripheral nerve injury can provide a key target for its clinical treatment. Currently, the role of Piezo1 in cortical pain processing is unclear. In the present study, we aimed to investigate the mechanism of Piezo1 expression in the ACC after peripheral nerve injury and its relationship with neuropathic pain. We found that SNI induced the abnormal expression of Piezo1 in ACC parvalbumin (PV)-expressing interneurons (PV-INs). The activation of the NLR family pyrin domain containing 3 (NLRP3) inflammasome, a fundamental component of the innate immune system, may be the key factor mediating Piezo1 upregulation in SNI, promoting an imbalance between ACC excitation and inhibition by inducing the microglial phagocytosis of PV-INs and, thereby, facilitating spinal pain transmission.

## 2. Results

### 2.1. SNI Induced Piezo1 Overexpression Preferentially in ACC Inhibitory PV-INs

Consistent with our previous findings [34,45], unilateral SNI induced a significant decrease in ipsilateral paw withdrawal thresholds in the present study, indicating that the neuropathic pain model was successfully duplicated. Next, we compared the bilateral expression of ACC Piezo1 between SNI and sham rat groups and found that SNI significantly increased the level of Piezo1 on both sides at PO day 7 (Figure 1A; *p*-value < 0.05 ipsilaterally, *p*-value < 0.01 contralaterally). Dual immunolabeling was used to identify the cells responsible for Piezo1 upregulation in the ACC. Piezo1 was mainly colocalized with NeuN (neuronal marker), but not with Iba1 (microglial marker) and GFAP (astrocyte marker), 7 d after SNI (Figure 1B–D). Hendrickx et al. suggested that Iba1, a marker of early microglial activation, is involved in microglial mobility and phagocytosis [46]. As shown by the arrow in Figure 1C, the signals for some Iba1-labeled microglia closely surrounded the Piezo1 signal, or even partially overlapped with it, suggesting that microglia can phagocytose Piezo1-immunoreactivity (IR) neurons.

To further confirm the neuronal profile of Piezo1 expression, we analyzed the co-localization of Piezo1 with excitatory glutaminergic neurons and major inhibitory parvalbumin-expressing interneurons (PV-INs) in the ACC. As shown in Figure 2, a large proportion of Piezo1-IR neurons was co-localized with inhibitory PV-IR neurons but not with glutaminergic CaMKⅡ-IR neurons.

### 2.2. SNI Induced the Loss of PV-INs and Inhibitory Synaptic Terminals, Potentially through Their Phagocytosis by Microglia

Synapses terminating at pyramidal neurons in the cerebral cortex are mostly gamma-aminobutyric acid (GABA)-ergic inhibitory synapses [47]. We next examined the number of PV-INs and the expression of the somatic inhibitory synaptic terminal marker vGAT. As shown in Figure 3, there was a reduction in the number of PV-INs but not in the number of CaMKⅡ positive neurons in the bilateral ACC in the SNI group 7 d after surgery compared with the sham group (*p*-value < 0.05). A significant loss in the number of vGAT positive puncta accompanied by microglial somatic adhesion in the bilateral ACC was also observed in the SNI group compared to the sham group (Figure 4A,B; *p*-value < 0.001 ipsilaterally, *p*-value < 0.01 contralaterally). The abovementioned changes induced by SNI were accompanied by corresponding pain behavior, which was reflected in decreased ipsilateral paw withdrawal thresholds at PO days 3, 5, and 7 (Figure 4C).

Combined with the encapsulation of Piezo1-positive neurons by Iba1^+^ microglia, as shown in Figure 1C, and the preferential expression of Piezo1 on PV-INs (Figure 2), it is speculated that the loss of PV-INs after SNI may be due to their phagocytosis by activated microglia. Further observation of microglial structure using the Iba1 marker (Figure 5A,B) showed that SNI activated the morphological transformation of microglia into amoeba-like cells, with a reduction of microglia process length (*p*-value < 0.05 ipsilaterally, *p*-value < 0.01 contralaterally) and an increase in cell body area (*p*-value < 0.001 bilaterally) in the ACC. CD68, a scavenger receptor, is another marker that is widely used to study the phagocytosis of microglia and mainly exists on intracellular lysosomal membranes [46]. As shown in Figure 5C, compared with the sham group, the fluorescence intensity of CD68 in the ACC was enhanced 7 days after SNI, especially in the contralateral side (*p*-value < 0.01). The immunoreactivity of CD68 partially overlapped with that of inhibitory synaptic endings vGAT or PV by double staining (Figure 5D).

### 2.3. Neuroinflammation in the ACC Is the Molecular Mechanism Underlying Piezo1 Overexpression Leading to Mechanical Allodynia after SNI

Our previous work [34] showed that SNI upregulated the protein levels of TNF-α in ACC neurons. NLRP3 inflammasome, a downstream factor of TNF-α [48,49], has been suggested as a key driving factor of many painful diseases and conditions [50]. The role of NLRP3 inflammasome in the mediation of the abnormal expression of Piezo1 has not been reported.

We first tested whether TNF-α directly induces NLRP3 inflammasome activation through cellular experiments. The administration of recombinant rat TNF-α (rrTNF) at a dose of 5 ng/mL into BV2 microglial cells elevated the protein levels of Piezo1 and NLRP3 (*p*-value < 0.05, Figure 6A). Similarly, the treatment of cultured BV2 cells with 5 ng/mL rrTNF also enhanced the fluorescence intensities of caspase-1 and interleukin-1β (IL-1β) (Figure 6B; *p*-value < 0.01 and *p*-value < 0.05, respectively), both of which are key downstream factors of activated NLRP3 inflammasome. Pretreatment of BV2 cells with the NLRP3 inhibitor MCC950 (10 nM) for 1 h blocked the rrTNF (5 ng/mL)-mediated induction of caspase-1 and IL-1β (Figure 6B), indicating that rrTNF activates NLRP3 inflammasome [48,49]. Furthermore, increased levels of Piezo1 and NLRP3 were also observed in the rrTNF-stimulated primary culture of ACC neurons (Figure 6C; *p*-value < 0.05 and *p*-value < 0.01, respectively). Treatment with the Piezo1 inhibitor GsMTx4 (1 nM) blocked the rrTNF-induced upregulation of Piezo1 but not that of NLRP3; the protein levels of NLRP3 in the rrTNF^+^/GsMTx4^+^ treatment group were still significantly higher than that of the control group (rrTNF^-^/GsMTx4^-^; Figure 6C; *p*-value < 0.05).

Further in vivo studies showed that SNI-induced NLRP3 colocalized with NeuN (neuronal marker), Iba1 (microglial marker) and GFAP (astrocyte marker), and activated microglia closely clung to NLRP3-IR cell bodies (Figure 7A). Almost all the PV-IR neurons were wrapped by the pore-forming protein, gasdermin D (GSDMD)-IR (Figure 7B). Blocking the activation of NLRP3 inflammasome using MCC950 (10 mg/kg i.p., once a day) reversed the SNI-induced overexpression of Piezo1 in the bilateral ACC (Figure 7C) and alleviated SNI-induced allodynia (Figure 7D), suggesting a potential mechanism underlying the abnormal expression of Piezo1 after SNI.

## 3. Discussion

In this study, we found that SNI increased the protein levels of Piezo1 in the bilateral ACC and was mainly localized to inhibitory PV-INs. A reduction in the numbers of PV-INs and inhibitory synaptic terminals was also observed in SNI rats. We also found that SNI activated the morphological transformation of microglia into amoeba-like cells with a reduced process length and increased cell body area in the ACC. Combined with the encapsulation of Piezo1-positive neurons by Iba1^+^ microglia, this finding indicates that the loss of PV-INs after SNI might result from their phagocytosis by the microglia. In cellular experiments, the treatment of BV2 cell culture or primary ACC neuron culture with rrTNF elevated the protein levels of Piezo1 and NLRP3. The administration of the NLRP3 inhibitor MCC950 to these cells blocked the rrTNF-mediated induction of caspase-1 and IL-1β in vitro, reversed the SNI-induced Piezo1 overexpression in the ACC, and alleviated SNI-induced allodynia in vivo.

### 3.1. SNI Increases the Expression of Piezo1 in ACC PV-INs

Central GABAergic interneurons are believed to play an important role in maintaining the excitatory and inhibitory (E/I) balance [51], and their dysfunction in the ACC is responsible for chronic inflammatory and neuropathic pain [52,53,54,55]. GABAergic interneurons are highly heterogeneous and have been classified based on several specific molecular markers: those expressing the Ca^2+^-binding protein parvalbumin (PV), including chandelier and basket cells; those expressing the neuropeptide somatostatin (SST or SOM); and those expressing the ionotropic serotonin receptor 5HT3a (5HT3aR), including vasoactive intestinal peptide (VIP)-positive and VIP-negative cells [56,57,58]. Each of these cell types has a different morphology, electrophysiology, connectivity, and pattern of in vivo activity [58]. Growing evidence indicates that fast- and non-adapting-spiking PV-INs are powerful regulators of the E/I balance and inhibit pyramidal neuron activity [59] by targeting the axonal initial segment (AIS, chandelier cells) or the perisomatic region and dendrites (basket cells) to drive changes in brain function, thus leading to behavioral responses [60,61].

Recent studies have found that SNI induces PV loss and AIS length reduction in the infralimbic cortex [62]; however, whether SNI causes such structural plasticity in the ACC has remained unclear. Pharmacological or chemogenetic activation of PV-INs, but not SOM-INs, in the ACC was revealed to alleviate mechanical allodynia and anxiety-like behavior in rats with chronic inflammatory pain [63,64,65]. PV-IR neurons in the human ACC comprise dendritic and axonal processes, and there are two dense PV-immunolabeled neuropil bands in ACC layers III and V [66]. PV-IR-positive axon cartridges are mainly located in ACC layers V and VI, and PV-IR protrusions in layer III can be traced extending vertically deep into layer V [67]. Combined with the knowledge that nerve-injury-induced neuronal hyperexcitability in ACC layers II and III [41,42,43] increases the firing of ACC corticospinal projection neurons in layers V and VI [67] through long-term potentiation (LTP) of synaptic transmission [35,68], the abovementioned findings suggest that SNI surgery may trigger the inhibitory dysfunction of PV-INs connecting pyramidal neurons and the resulting E/I imbalance in the ACC deep layer, thus enhancing the descending facilitation of spinal pain transmission from the ACC [69]. 

Piezo1 is a mechanosensitive ion channel that can be activated by cytokines associated with neuroinflammatory pathology and lipopolysaccharide (LPS), an inflammatory stimulator [70,71]. Activated Piezo1 triggers Ca^2+^ influx and Ca^2+^ release from intracellular stores [26,27,28], which can cause the pathophysiological production of reactive oxygen species (ROS) and oxidative damage [72,73,74]. Abnormally high Ca^2+^ influx also leads to the instability and unwinding of axonal cytoskeletal transport mechanisms, ultimately leading to irreversible neurodegeneration [75,76]. Therefore, the disturbance of the immune cytokine microenvironment in the ACC after SNI [34,45] is likely to cause abnormal Piezo1 expression and, thereby, contribute to neuropathic pain. In the present study, we found that the expression of Piezo1 mechanical channels was specifically increased in ACC PV-INs after SNI (Figure 1 and Figure 2). A reduction in the number of PV-INs and a decrease in the number of somatic inhibitory axon terminals in the ACC was observed in rats with neuropathic pain (Figure 3 and Figure 4). Our findings confirm, for the first time, that SNI increases the expression of Piezo1 specifically in ACC PV-INs, which might be the main factor driving PV-INs loss and E/I imbalance in the ACC.

### 3.2. NLRP3 Induces the Microglial Phagocytosis of PV-INs by Enhancing Piezo1 Expression

The microglia are the primary resident immune cells in the cortical nervous system, influencing brain development through monitoring, scavenging, and synaptic pruning, and participate in CNS injury and repair [77,78,79,80]. Under physiological conditions, the microglial morphology is ramified; in case of injury or neuroinflammation, the microglia change to take an amoeba-like morphology conferring an increased phagocytosis ability [77,81]. Microglial activation and PV-IN dysfunction in the hippocampus and medial prefrontal cortex (mPFC) were achieved in the systemic inflammatory model prepared using intraperitoneal lipopolysaccharide (LPS) injection [82,83,84]. Activated microglia may promote central disinhibition by selectively phagocytosing inhibitory synaptic terminals [84] or displacing inhibitory axosomatic presynaptic terminals from cortical neurons [85]; moreover, blocking microglial activation alleviates PV-IN dysfunction and behavioral changes caused by inflammation or brain injury [82,83,86]. In addition, activated microglia may also interfere with the activity of inhibitory synapses by degrading perineuronal nets (PNNs) [87,88], a specialized extracellular matrix structure enwrapping cortical inhibitory fast-spiking PV-INs [89,90]. Our recent work demonstrated that SNI induces the activation of microglia, but not astrocytes, in the bilateral ACC [34,45]. However, whether the SNI-induced decrease in the number of PV-INs in the ACC is due to microglial phagocytosis remained unclear. The present study showed that activated ACC microglia induced by SNI (Figure 5) adhere tightly to the cell bodies of Piezo1-IR neurons (Figure 1) that are colocalized specifically with PV-IR (Figure 2). As discussed above, Piezo1 may induce cytotoxicity and neurodegeneration by promoting intracellular calcium signal accumulation [72,73,74,75,76]. CD68 is responsible for the clearance of debris [91] and phagocytosis of apoptotic cells [92]. Here, we found that SNI elevated the expression of CD68 in ACC and the immunoreactivity of CD68 partially overlapped with that of PV and inhibitory synaptic endings vGAT (Figure 5), suggesting that inhibitory PV-INs enable microglia to find them by increasing Piezo1 expression; this, in turn, reduces inhibitory innervation by engulfing or degrading inhibitory synaptic endings (Figure 4). 

Our previous study showed that SNI elevated the expression of TNF-α in contralateral ACC neurons, which are responsible for pain aversiveness and pain maintenance [34]. The role of TNF-α as a major regulator of balance between cell survival and death has been established [93]. The traditional view is that TNF-α induces an inflammatory response and programmed cell death through apoptosis and necroptosis according to different pathological conditions [94,95]. Of note, TNF-α also triggers the activation of NLRP3 inflammasome, an important immune-reaction-initiating complex that acts in combination with the adaptor molecule apoptosis-associated speck-like protein containing CARD (ASC), which controls the activation of caspase-1 and its subsequent inflammatory response as well as pyroptosis mediated by the pore-forming protein gasdermin D (GSDMD) [48,49]. Reducing spinal inflammation and pyroptosis by targeting NLRP3 alleviates chronic constriction injury (CCI)-induced allodynia and hyperalgesia [96]. NLRP3 inflammasome is also involved in the cortical neuroinflammation and neurodegenerative changes associated with trigeminal neuralgia [97] and migraine [98]. Interestingly, a recent study found that the activation of Piezo1 in nucleus pulposus cells promoted NLRP3 inflammasome assembly through the Ca^2+^/ NF-κB pathway [99]. Although most studies have shown that NLRP3 inflammasome is assembled inside the microglia [97,100], NLRP3 inflammasome assembly and subsequent pyroptosis can also be observed in cortical neurons [98]. However, in the present study, the administration of the Piezo1 inhibitor GsMTx4 to primary ACC neurons blocked the rrTNF-stimulated upregulation of Piezo1 but not NLRP3 (Figure 6C), indicating that Piezo1 cannot activate NLRP3 inflammasome.

Studies have shown that the expression levels of NLRP3 and IL-1β are elevated in several rodent neuropathic pain models [50,101]. But one team reported that SNI could not alter the expression of spinal NLRP3 inflammasome components (including IL-1β) [102], contradicting the results reported by others that SNI increased the expression of IL-1β in the serum, cerebrospinal fluid [103], and spinal cord [104,105]. In the present work, the administration of rrTNF to cultured BV2 cells or primary ACC neurons elevated the protein levels of Piezo1 and NLRP3 (Figure 6). Pretreatment with the NLRP3 inhibitor MCC950 blocked the rrTNF-stimulated induction of caspase-1 and IL-1β in vitro (Figure 6), reversed the SNI-induced Piezo1 overexpression in the ACC, and alleviated SNI-induced allodynia in vivo (Figure 7), suggesting that the SNI-induced overexpression of TNF-α in ACC [34] may be the cause of Piezo1 upregulation via NLRP3 inflammasome activation.

## 4. Materials and Methods

### 4.1. Animals

The animals used in this study were adult male Sprague-Dawley rats (160–200 g) and lived in individual cages with strictly controlled humidity (50–60%), temperature (24 °C), and 12 h light/dark cycle (06:00 h–18:00 h). Food and water were made freely available.

### 4.2. Spared (Sciatic) Nerve Injury (SNI) 

SNI surgical operations were performed as previously described by Decosterd et al. [44]. First, the animals were anesthetized by intraperitoneal injection (i.p.) of 0.4% sodium pentobarbital (40 mg/kg, Sigma-Aldrich, Taufkirchen, Germany). Then, we incised the skin on the outer surface of the left thigh to expose the sciatic nerve and its three terminal branches: the sural nerve, the tibial nerve, and the common peroneal nerve. Finally, the peroneal and tibial nerves were ligated and excised by 2 mm, leaving the peroneal nerve unharmed. The same exposure procedure was performed on the uninjured nerves in the sham group for comparison.

### 4.3. Mechanical Allodynia Test

Mechanical allodynia of the hind paws of rats was measured by the up and down movement method of Von Frey hair [106]. In short, animals were housed in individual plexiglass chambers on a net table. After 15 min of habituation, the allodynia test was conducted. Von Frey hairs (0.41, 0.70, 1.20, 2.04, 3.63, 5.50, 8.51, and 15.14 g) with logarithmically increasing hardness were applied bilaterally to the hind paw starting from 2.04 g. The 50% paw withdrawal threshold was recorded, and the reaction to mechanical stimulation was evaluated at different postoperative (PO) times.

### 4.4. Immunohistochemistry and Immunofluorescence

After being anesthetized, the ascending aorta was perfused with 0.9% saline followed by 4% paraformaldehyde prepared with 0.1 M phosphate buffer (PB). After perfusion, the brain tissue was removed and fixed for 5 h and dehydrated with 30% sucrose for 5 days. Coronal sections (25 to 35 µm thick) were then made from brain tissue (bregma +3.0 to +1.7 mm) containing ACC using a cryosectioner (LEICA CM3050S, Wetzlar, Germany). Next, the cells cultured on coverslips were washed three times and then fixed in 4% paraformaldehyde for 30 min at room temperature. Paraformaldehyde was discarded, and the cells were then washed three times for 5 min each.

For immunohistochemistry, prior to incubation with primary antibodies, the sections or cells were treated with 5% donkey serum for 1 h at room temperature. After overnight incubation with primary antibody at 4 °C, the sections or cells were washed with PBS and incubated for 1 h with secondary antibody at room temperature.

For the double staining, sections were incubated with a mixture of primary antibodies derived from different species. Incubation with anti-Piezo1 antibody (1:100, Alomone, Jerusalem, Israel) plus either anti-Parvalbumin antibody (1:500, NOVUS, Littleton, CO, USA), anti-CaMKⅡ recombinant rabbit monoclonal antibody (1:200, HuaBio, Zhejiang, China), anti-CD68 antibody (1:200; Abcam, Cambridge, UK) or anti-NLRP3 antibody (1:100, Affinity, Cincinnati, OH, USA) was performed overnight at 4 °C. Brain slices were stained with anti-CD68 antibody and anti-Parvalbumin antibody. Incubation with anti-NLRP3 antibody and anti-NeuN (neuronal marker, 1:200; CST, Danvers, MA, USA), anti-GFAP (astrocyte marker, 1:400; CST), or anti-Iba1 (microglia marker, 1:500; Abcam, Cambridge, UK) was also performed. The cells were double-stained with anti-IL-1β antibody (1:200, ABclonal, Woburn, MA, USA) and anti-caspase-1 antibody (1:200, ABclonal, Woburn, MA, USA). After three washes with PBS (10 min each), the sections were incubated with Donkey Anti-Goat IgG H&L conjugated with Alexa Fluor^®®^ 647 (1:400, Abcam, Cambridge, UK) and fluorescein isothiocyanate (FITC)- and Cy3-conjugated secondary antibodies (1:400, Jackson Immuno Research, Philadelphia, PA, USA) for 1 h at room temperature (24–26 °C). 

Images were captured using a fluorescent microscope attached to a CCD point camera (Leica DFC350FX/DMIRB, Heidelberg, Germany) and the Leica-IM50 software. For vGAT fluorescence signal, images were acquired with a Nikon confocal microscope (C2+, Nikon, Tokyo, Japan) by using a 60× (oil) objective. We used ImageJ (National Institutes of Health, Bethesda, MD, USA) or NIS-element to measure signal intensity. In order to confirm the specificity of primary antibody immunostaining, negative control sections without primary antibodies were processed in parallel.

### 4.5. Western Blotting

Rats were decapitated and their ACC were removed. Next, we centrifuged tissue samples at 12,000× *g* for 20 min at 4 °C and quantified target proteins. Gel electrophoresis (SDS-PAGE) was used to separate proteins, which were then electro-transferred to PVDF membranes (Millipore, Billerica, MA USA). After being sealed with 5% skimmed milk at room temperature for 1 h, the membranes were placed in anti-NLRP3 antibody (1:1000, Affinity, Cincinnati, OH, USA) and anti-Piezo1 antibody (1:1000, Alomone, Jerusalem, Israel) overnight at 4 °C and then incubated with HRP-conjugated donkey anti-mouse, anti-goat, or anti-rabbit secondary antibody (1:10,000, Abcam, Cambridge, UK). β-actin (1:1000, Boster, Wuhan, China) was used as a load control. Enhanced chemiluminescence (Bio-Rad, Hercules, CA, USA) was used to detect the target protein bands, which were imaged using the Tanon-5200 chemiluminescence imaging system (Tanon Science and Technology, Shanghai, China). An image analysis system (KONTRON IBAS 2.0, Germany) was used to quantify the protein level in comparison to the expression level of β-actin.

### 4.6. BV2 Cell Line and Culture

BV2 cells obtained from Pain Research Center were used for Western blotting and immunohistochemistry experiments. The cells were cultured in Dulbecco’s Modified Eagle Medium (DMEM, GIBCO, Berlin, USA) supplemented with 10% fetal bovine serum (FBS, GIBCO, Darmstadt, Germany) and 1% antibiotic-antimycotic (GIBCO, Berlin, Germany), and the medium was changed every 2 days. Experimental and control groups were treated with tumor necrosis factor α (TNF-α, Novoprotein, Jiangsu, China) at various concentrations and PBS simultaneously.

### 4.7. Primary Culture of ACC Neurons from Neonatal Rats

ACC neurons were dissected from neonatal rat (within 1–2 days after birth) using the neonatal rat brain atlas. Briefly, during anesthesia with isoflurane (RWD, Shenzhen, China), the rats were decapitated and then euthanized. Next, the ACC areas of the brains were isolated and dissociated in 2 g/L papainase and 100 µg/L DNase (Sigma-Aldrich, Taufkirchen, Germany) for 30 min at 37 °C and then centrifuged at 1000 rpm for 5 min. After that, the supernatant was discarded and the sample was resuspended in DMEM-F12 (GIBCO, Darmstadt, Germany)) with 10% FBS and 100 µg/L DNase; it was then allowed to rest for 2 min. The single cell suspension was collected and the undigested tissue pieces were put through the above process two more times. Finally, to remove any remaining tissue fragments, the single cell suspensions were filtered through a mesh of 70 microns and were plated on six-well plates (NEST, Jiangsu, China) coated with poly-d-lysine (0.2 mg/mL, Sigma-Aldrich, Taufkirchen, Germany) at a density of 1 × 10^8^ cells/L in DMEM-F12 with 10% FBS, 1% antibiotic-antimycotic, and 2 mM L-Glutamine (Sigma-Aldrich, Taufkirchen, Germany) at 37 °C in a humidified incubator containing 5% CO_2_. Then, 24 h after plating, the medium was replaced with Neurobasal-A Medium (GIBCO) containing 10% FBS, 2 mM L-glutamine, 1% penicillin:streptomycin, and 2% B-27 solution (GIBCO); the medium was changed every 2–3 days thereafter. The experiments were commenced once the neurons were stable.

### 4.8. Intra-ACC Drug Application

Stereotaxic surgery was performed according to the rat brain atlas. A stainless steel guide cannula with a stainless steel pin plug was inserted into the ACC contralateral to the nerve injury and secured to the skull with acrylic dental cement. The stereoscopic coordinates of the ACC injection site were as follows: AP +1.5 mm, ML +0.5 mm, DV −2.5 mm (from bregma). The following experiments were initiated 1 week after recovery. GsTMx4, an inhibitor of cationic mechanosensitive channels, was slowly delivered into the ACC (10 mM, 2 μL, GlpBio, Montclair, CA, USA) within 5~10 min. Starting one day before SNI surgery, GsTMx4 or vehicle was taken daily for 8 days; a sterile saline solution was administered to the control groups instead.

### 4.9. Statistical Analysis

The statistical analysis for immunohistochemistry was performed as described previously [34]. GraphPad Software was used for multiple *t*-test or unpaired *t*-test or one-way ANOVA or two-way ANOVA to determine the differences over different groups where appropriate. For behavioral data testing, Dunn’s multiple comparisons test or nonparametric two-way ANOVA followed by the *Friedman* test were performed. In all cases, the results are presented as mean ± SEM and statistical significance was considered to be *p* < 0.05.

## 5. Conclusions

Our previous work has shown that SNI triggers ACC hyperexcitability [34] and that control of the ACC chemokine-mediated immune cascade alleviates SNI-induced spinal neuroinflammation and mechanical allodynia [45]. In this work, at a molecular level, we further researched NLRP3 inflammasome in ACC PV-INs following SNI-mediated Piezo1 upregulation. Piezo1 induces cell damage by intracellular calcium accumulation, which enables microglial phagocytosis, leading to the reduction of PV-INs. Secondarily, the ACC E/I imbalance disinhibits pyramidal neurons and, subsequently, facilitates spinal pain transmission. Strategies that target the NLRP3-Piezo1 pathway in ACC PV-INs may act as novel therapeutic modalities for chronic pain.

## Figures and Tables

**Figure 1 ijms-23-13035-f001:**
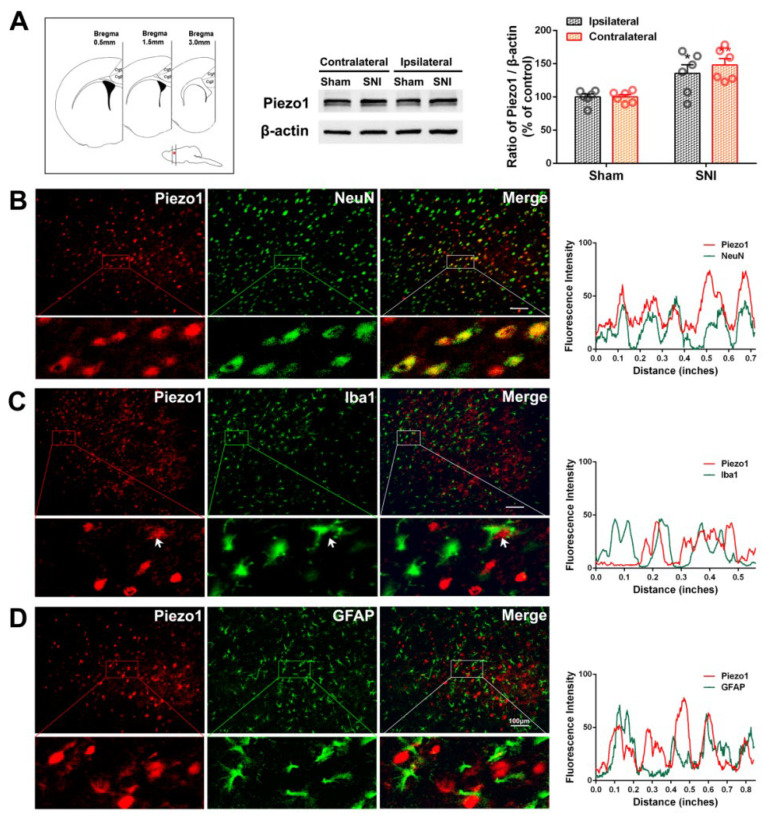
SNI induces Piezo1 overexpression in ACC neurons. (**A**) SNI-induced bilateral expression of Piezo1 in the ACC. Left, Cg1 and Cg2 tissues were used for Western blotting. Coronal dissection was performed 3.0–0.5 mm anterior to the Bregma. A representative Western blot of Piezo1 expression in the bilateral ACC is shown in the middle. The protein quantification results from Western blotting are shown on the right. Significant differences in Piezo1 expression were observed on both sides of the ACC 7 d after SNI. * *p*-value < 0.05, ** *p*-value < 0.01 versus the sham group (two-way ANOVA). (**B**–**D**) Representative double immunofluorescence staining showing the overlap (yellow) of Piezo1-IR (red) with neuronal marker NeuN (green), but not with microglial marker Iba1 (green) and astrocyte marker GFAP (green), 7 d after SNI. The arrowhead in (**C**) indicates Iba1-IR surrounding or partially overlapping Piezo1-IR. Scale bar = 50 μm. The fluorescence intensity curves for red and green signals in the boxed areas are shown on the right side for each group.

**Figure 2 ijms-23-13035-f002:**
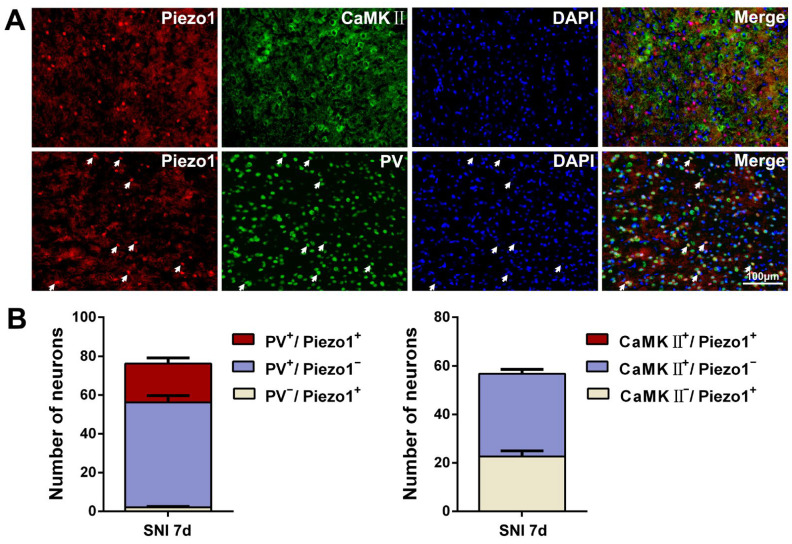
SNI-induced Piezo1 is preferentially expressed in ACC PV-INs. (**A**) Representative double staining image showing the overlap (white arrows) of Piezo1-IR (red) with PV (inhibitory interneuron marker, green, below), but not with CaMK II (glutamate neuronal marker, green, top), 7 d after SNI. The color of co-localization is yellow. Blue fluorescence is from DAPI, a nuclear counterstain. Scale bar = 100 μm. (**B**) Quantitative analysis of Piezo1-IR neurons co-localized with PV-IR and CaMKⅡ-IR neurons.

**Figure 3 ijms-23-13035-f003:**
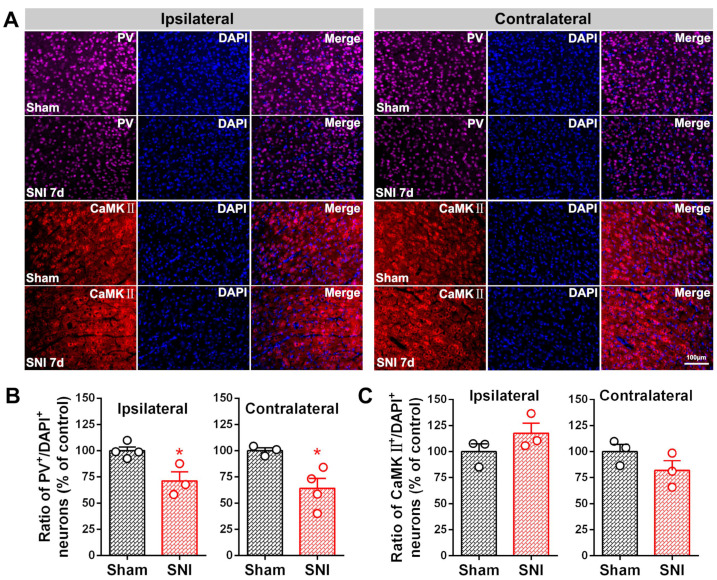
SNI triggers a reduction in the number of ACC inhibitory PV-INs. (**A**) Representative immunofluorescence staining image showing PV-IR (magenta) and CaMKⅡ-IR (red) neurons in the bilateral ACC in both sham and SNI rat groups. Blue fluorescence is from DAPI, a nuclear counterstain. Scale bar = 100 μm. Quantitative analysis for the percentages of (**B**) PV-IR (PV^+^) and (**C**) CaMKⅡ-IR (CaMKⅡ^+^) neurons. * *p*-value < 0.05 versus the sham (unpaired *t* test).

**Figure 4 ijms-23-13035-f004:**
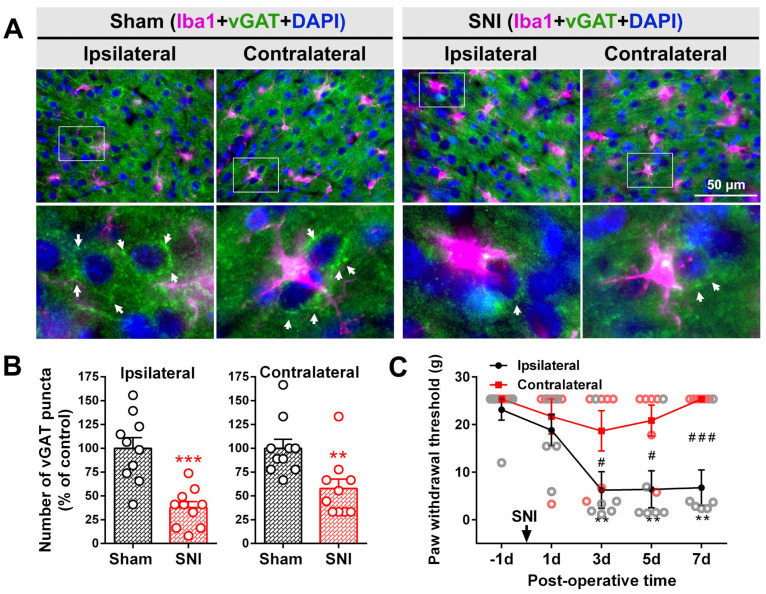
SNI triggers the loss of inhibitory synaptic terminals in the bilateral ACC. (**A**) Representative double immunofluorescence staining image showing somatic vGAT puncta-IR (inhibitory synaptic terminal marker, green) in the sham and SNI groups and the somatic adhesion of Iba1-IR (microglial marker, magenta) in the bilateral ACC. Scale bar = 50 μm. The enlarged image in the white box is shown below and the white arrow shows the vGAT puncta signal. Blue fluorescence is from DAPI, a nuclear counterstain. (**B**) Comparison of the number of vGAT-IR puncta in the bilateral ACC of sham and SNI groups. ** *p*-value < 0.01, *** *p*-value < 0.001 versus the sham (unpaired *t* test). (**C**) Changes in the bilateral mechanical paw withdrawal thresholds of SNI rats. Significant differences were observed in the threshold for the ipsilateral paw on PO days 3, 5, and 7 (*n* = 6). ** *p*-value < 0.01 versus PO day −1 (Dunn’s multiple comparisons test) or # *p*-value < 0.05, ### *p*-value < 0.001 versus the contralateral side (multiple *t* test).

**Figure 5 ijms-23-13035-f005:**
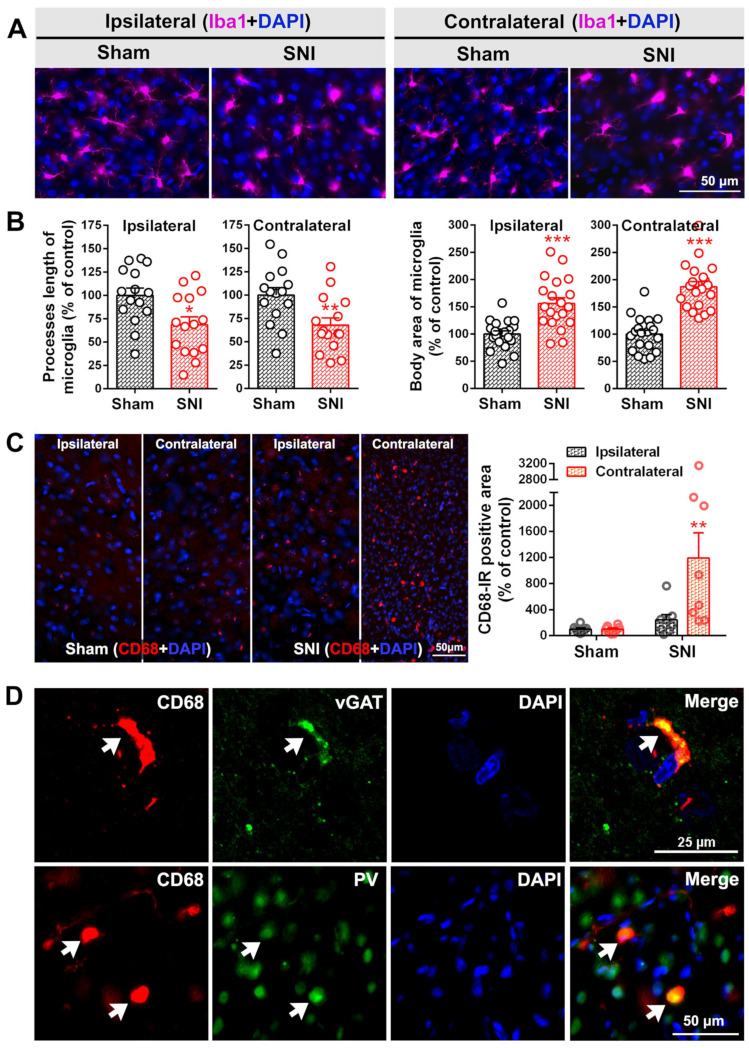
SNI activates the morphological transformation of microglia into amoeba-like cells. (**A**) Representative immunofluorescence staining image of Iba1-IR (microglial marker, magenta) in the bilateral ACC of sham and SNI groups. Scale bar = 50 μm. (**B**) Comparison of the process lengths and cell body areas of microglia in the bilateral ACC of sham and SNI groups. * *p*-value < 0.05, ** *p*-value < 0.01, *** *p*-value < 0.001 versus the sham group (unpaired *t* test). (**C**) Left: representative immunofluorescence staining image showing CD68-IR in the bilateral ACC in both sham and SNI rats. Scale bar = 50 μm. Right: quantification of the CD68 fluorescence signals in both groups. ** *p*-value < 0.01 versus the sham group (two-way ANOVA). (**D**) White arrow heads indicate the co-localization of CD68-IR (red) with that of vGAT (green, top) and PV (green, below) 7d post SNI by double staining. Scale bar = 25 μm in the top line; 50 μm in the bottom line. Blue fluorescence in (A, C and D) is from DAPI, a nuclear counterstain.

**Figure 6 ijms-23-13035-f006:**
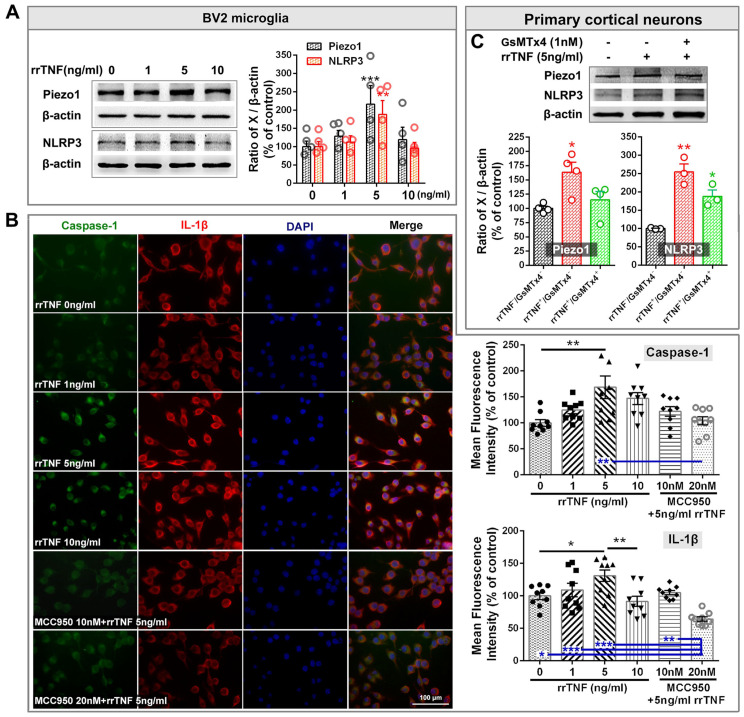
The effect of recombinant rat TNF-α (rrTNF) on the expression levels of Piezo1 and NLRP3 in cellular experiments. (**A**) A representative Western blot showing Piezo1 and NLRP3 protein levels after the treatment of BV2 cells with different doses of rrTNF (0, 1, 5, and 10 ng/mL) is presented on the left. The protein quantifications are shown on the right. ** *p*-value < 0.01, *** *p*-value < 0.001 versus the control (0 ng/mL) group (two-way ANOVA). (**B**) Left: Representative double staining showing changes in the immunofluorescence intensities of caspase-1 (green) and IL-1β (red) following treatment of cultured BV2 cells with different doses of rrTNF (0, 1, 5, and 10 ng/mL) and the effect of the NLRP3 inhibitor MCC950 on the rrTNF-mediated induction of caspase-1 (green) and IL-1β in BV2 cells. Blue fluorescence is from DAPI. Scale bar = 50 μm. Right: Quantitative analysis of the fluorescence intensities of caspase-1 and IL-1β in the different groups. * *p*-value < 0.05, ** *p*-value < 0.01 (one-way ANOVA). (**C**) Representative Western blot showing Piezo1 and NLRP3 protein levels following the treatment of the primary ACC neuron culture with rrTNF (5 ng/mL) and the effect of the Piezo1 inhibitor GsMTx4 on these rrTNF-mediated inductions are shown on the top. The quantification results are presented below. * *p*-value < 0.05, ** *p*-value < 0.01, *** *p*-value < 0.001 versus the control (one-way ANOVA).

**Figure 7 ijms-23-13035-f007:**
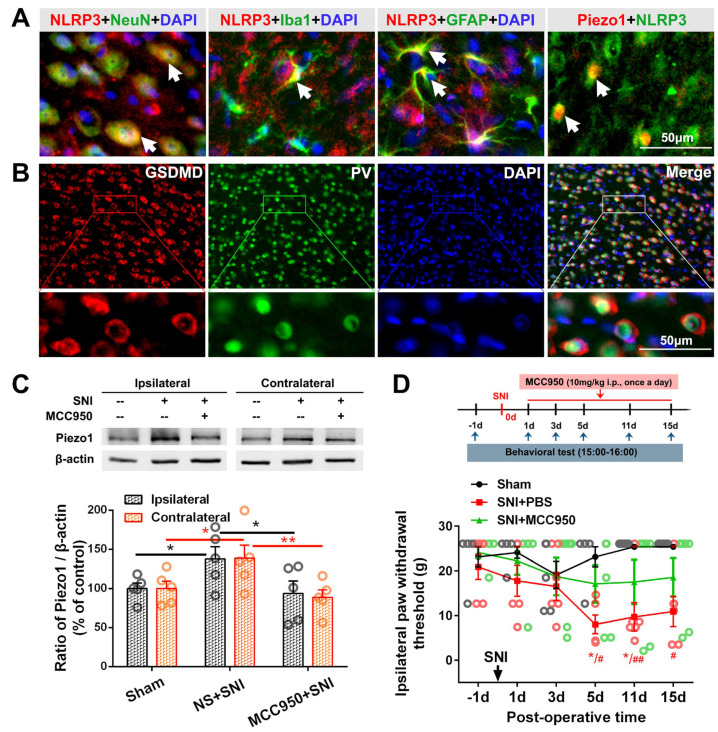
The effect of NLRP3 inhibition on SNI-induced Piezo1 upregulation and pain behavior. (**A**) Representative double staining showing the overlap of NLRP3-IR (red) with NeuN, Iba1 and GFAP (green), 7 d after SNI. NLRP3-IR (green) also colocalized with Piezo1-IR neurons (red). White arrow shows the co-localization (yellow). Scale bar = 50 μm. (**B**) Representative double immunofluorescence staining image showing the encapsulation of PV-IR by GSDMD-IR, a pore-forming protein downstream to NLRP3 inflammasome. Scale bar = 50 μm. Blue fluorescence in (A and B) is from DAPI. (**C**) Representative Western blot showing the effect of the NLRP3 inhibitor MCC950 (10 mg/kg i.p., once a day for 7 days) on the Piezo1 protein level in the bilateral ACC (top). Protein quantification results (bottom). * *p*-value < 0.05, ** *p*-value < 0.01 (two-way ANOVA). (**D**) Changes in ipsilateral paw withdrawal thresholds in sham, SNI + PBS, and SNI + MCC950 group. Significant differences in the threshold were observed in SNI + PBS rats but not SNI + MCC950 rats compared with sham rats on PO days 5, 11 and 15 (*n* = 6). * *p*-value < 0.05 versus PO day −1 (Dunn’s multiple comparisons test) or # *p*-value < 0.05, ## *p*-value < 0.01 versus the sham group (multiple *t* test).

## Data Availability

All the necessary data are included within the article. Further data will be shared by request.

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
