# Peer review of "NLRP3-Mediated Piezo1 Upregulation in ACC Inhibitory Parvalbumin-Expressing Interneurons Is Involved in Pain Processing after Peripheral Nerve Injury"

_ijms, 2022, doi:10.3390/ijms232113035_

Round 1

Reviewer 1 Report

Congratulations to the authors on their wonderful work! I really really liked this article.

The topic is very relevant because it is devoted to the search for new biological mechanisms of the formation of chronic pain syndrome and the development of new therapeutic strategies. The article is well structured, the figures significantly improve the understanding of the results obtained. I would especially like to mention the extensive Discussion section.

I propose to make minor changes:

1) add initials and contact phone number to the information about the corresponding author;

2) complete the Introduction section with the stated purpose of this study;

3) try to avoid unnecessary abbreviations in the name of subsection 2.1;

4) replace p with p-value;

5) add the full name of all the numerous abbreviations at their first mention (for example, GsMTx4, CX3CL1, NLR, etc.) in the text and in the legends of all figures.

Author Response

Congratulations to the authors on their wonderful work! I really really liked this article.

The topic is very relevant because it is devoted to the search for new biological mechanisms of the formation of chronic pain syndrome and the development of new therapeutic strategies. The article is well structured, the figures significantly improve the understanding of the results obtained. I would especially like to mention the extensive Discussion section.

I propose to make minor changes:

Point 1: add initials and contact phone number to the information about the corresponding author;

Response 1: Thanks for your suggestion. We have added them in the revised manuscript.

Point 2:  complete the Introduction section with the stated purpose of this study;

Response 2: Thanks for your suggestion. “we aimed to investigate the mechanism of Piezo1 expression in ACC after peripheral nerve injury and its relationship with neuropathic pain.” was added on page 3.

Point 3: try to avoid unnecessary abbreviations in the name of subsection 2.1;

Response 3: Thanks for your comments. We have deleted unnecessary abbreviations on page 3-6.

Point 4: replace p with p-value;

Response 4: Thanks for your suggestion. We have replaced “p” with “p-value” in the results section in the revised manuscript.

Point 5: add the full name of all the numerous abbreviations at their first mention (for example, GsMTx4, CX3CL1, NLR, etc.) in the text and in the legends of all figures.

Response 5: Thank you. We have thoroughly revised the manuscript and added them in the text and the legends of all figures.

Reviewer 2 Report

This study provides novel molecular information regulated to pain processing. Although the merit of some of the data is valued, this reviewer feels that the image on Figure 1C does not support overlapping Piezo-1 Iba1 signals to suggest microglia phagocytosis of Piezo-1 positive neurons. Could the authors provide stronger evidence for this claim which seems key in the proposed mechanism?

Minor comments:

It seems that Line 57 should read Piezo 1 insteado of “Piezo 2”.

Line 66 it should read Piezo 1 instead of “Peizo 1”.

Author Response

Point 1: This study provides novel molecular information regulated to pain processing. Although the merit of some of the data is valued, this reviewer feels that the image on Figure 1C does not support overlapping Piezo-1 Iba1 signals to suggest microglia phagocytosis of Piezo-1 positive neurons. Could the authors provide stronger evidence for this claim which seems key in the proposed mechanism?

Response 1: Thanks for your comments. This sentence of “Hendrickx et al. suggested that Iba1, a marker of early microglial activation, is involved in microglial mobility and phagocytosis [46].” is added to the revised manuscript on lines 102 to 103.

To provide stronger evidence that microglia engulf PV-INs, we added a set of experiments, and the experimental results are presented as Figure 5C and D in the revised manuscript, and the relevant descriptions “CD68, a scavenger receptor, is another marker that's widely used to study the phagocytosis of microglia and mainly exists on intracellular lysosomal membranes [46]. As shown in Figure 5C, compared with the sham group, the fluorescence intensity of CD68 in ACC was enhanced 7 days after SNI, especially in the contralateral side (p-value < 0.01). The immunoreactivity of CD68 partially overlapped with that of inhibitory synaptic endings vGAT or PV by double staining (Figure 5D).” is presented on lines 167 to 172.

This paragraph of “As discussed above, Piezo1 may induce cytotoxicity and neurodegeneration by promoting intracellular calcium signal accumulation [72-76]. CD68 is responsible for the clearance of debris [91] and phagocytosis of apoptotic cells [92]. Here, we found that SNI elevated the expression of CD68 in ACC and the immunoreactivity of CD68 partially overlapped with that of PV and inhibitory synaptic endings vGAT (Figure 5), ” is added in lines 324 to 329.

Added Reference in revised manuscript:

[46] Hendrickx, D. A. E.; van Eden, C. G.; Schuurman, K. G.; Hamann, J.; Huitinga, I., Staining of HLA-DR, Iba1 and CD68 in human microglia reveals partially overlapping expression depending on cellular morphology and pathology. J Neuroimmunol 2017, 309, 12-22.

[91] Hendrickx, D. A.; Koning, N.; Schuurman, K. G.; van Strien, M. E.; van Eden, C. G.; Hamann, J.; Huitinga, I., Selective upregulation of scavenger receptors in and around demyelinating areas in multiple sclerosis. J Neuropathol Exp Neurol 2013, 72, 106-118.

[92] Chistiakov, D. A.; Killingsworth, M. C.; Myasoedova, V. A.; Orekhov, A. N.; Bobryshev, Y. V., CD68/macrosialin: not just a histochemical marker. Laboratory Investigation 2017, 97, 4-13.

Minor comments:

Point 2: It seems that Line 57 should read Piezo 1 instead of “Piezo 2”.

Response 2: The Piezo channel subtypes described in the first manuscript (line 57-60) “Notably, although Piezo2 is expressed in the vast majority (≥ 90%) of DRG neurons, its expression in the brain is limited to select neuron types in specific regions, including neocortical and hippocampal pyramidal neurons, cerebellar Purkinje cells, and olfactory bulb mitral cells.” is Piezo2 rather than Piezo1, which is why we choose to investigate the role of Piezo1 in ACC rather than that of Piezo2 in this work.

Point 3: Line 66 it should read Piezo 1 instead of “Peizo 1”.

Response 3: Thank you. We apologize for the error and have now made the appropriate change.

Round 2

Reviewer 2 Report

Raised concerns were properly addressed.